# UNSUPERVISED VIDEO-TO-VIDEO TRANSLATION

## ABSTRACT

Unsupervised image-to-image translation is a recently proposed task of translating an image to a different style or domain given only unpaired image examples at training time. In this paper, we formulate a new task of unsupervised video-to-video translation, which poses its own unique challenges. Translating video implies learning not only the appearance of objects and scenes but also realistic motion and transitions between consecutive frames. We investigate the performance of per-frame video-to-video translation using existing image-to-image translation networks, and propose a spatio-temporal 3D translator as an alternative solution to this problem. We evaluate our 3D method on multiple synthetic datasets, such as moving colorized digits, as well as the realistic segmentation-to-video GTA dataset and a new CT-to-MRI volumetric images translation dataset. Our results show that frame-wise translation produces realistic results on a single frame level but underperforms significantly on the scale of the whole video compared to our three-dimensional translation approach, which is better able to learn the complex structure of video and motion and continuity of object appearance.

## 1 INTRODUCTION

Recent work on unsupervised image-to-image translation (Zhu et al., 2017; Liu and Tuzel, 2016; Liu et al., 2017b) has shown astonishing results on tasks like style transfer, aerial photo to map translation, day-to-night photo translation, unsupervised semantic image segmentation and others. Such methods learn from unpaired examples, avoiding tedious data alignment by humans. In this paper, we propose a new task of unsupervised video-to-video translation, *i.e.* learning a mapping from one video domain to another while preserving high-level semantic information of the original video using large numbers of *unpaired* videos from both domains. Many computer vision tasks can be formulated as video-to-video translation, *e.g.*, semantic segmentation, video colorization or quality enhancement, or translating between MRI and CT volumetric data (illustrated in Fig. 1). Moreover, motion-centered tasks such as action recognition and tracking can greatly benefit from the development of robust unsupervised video-to-video translation methods that can be used out-of-the-box for domain adaptation.

Since a video can be viewed as a sequence of images, one natural approach is to use an image-to-image translation method on each frame, e.g., applying a state-of-art method such as CycleGAN (Zhu et al., 2017), CoGAN (Liu and Tuzel, 2016) or UNIT (Liu et al., 2017b). Unfortunately, these methods cannot preserve continuity and consistency of a video when applied frame-wise. For example, colorization of an object may have multiple correct solutions for a single input frame, since some objects such as cars can have different colors. Therefore, there is no guarantee that an object would preserve its color if translation is performed on the frame level frame.

In this paper, we propose to translate an entire video as a three-dimensional tensor to preserve its cross-frame consistency and spatio-temporal structure. We employ multiple datasets and metrics to evaluate the performance of our proposed video-to-video translation model. Our synthetic datasets include videos of moving digits of different colors and volumetric images of digits imitating medical scans. We also perform more realistic segmentation-to-RGB and colorization experiments on the GTA dataset (Richter et al., 2016), and propose a new MRI-to-CT dataset for medical volumetric image translation, which to our knowledge is the first open medical dataset for unsupervised volume-to-volume translation.

MRI→CT                  moving MNIST colorization

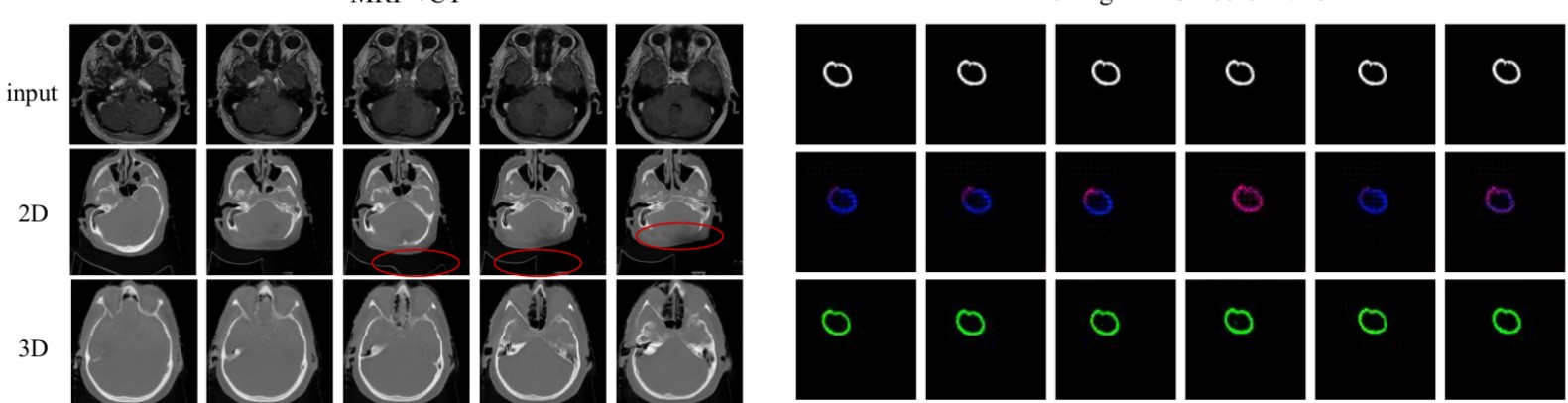

Figure 1: We propose the task of unsupervised video-to-video translation. Left: Results of MR-to-CT translation. Right: moving MNIST digits colorization. Rows show per-frame CycleGAN (2D) and our spatio-temporal extension (3D). Since CycleGAN takes into account information only from the current image, it produces reasonable results on the image level but fails to preserve the shape and color of an object throughout the video. *Best viewed in color.*

GTA colorization

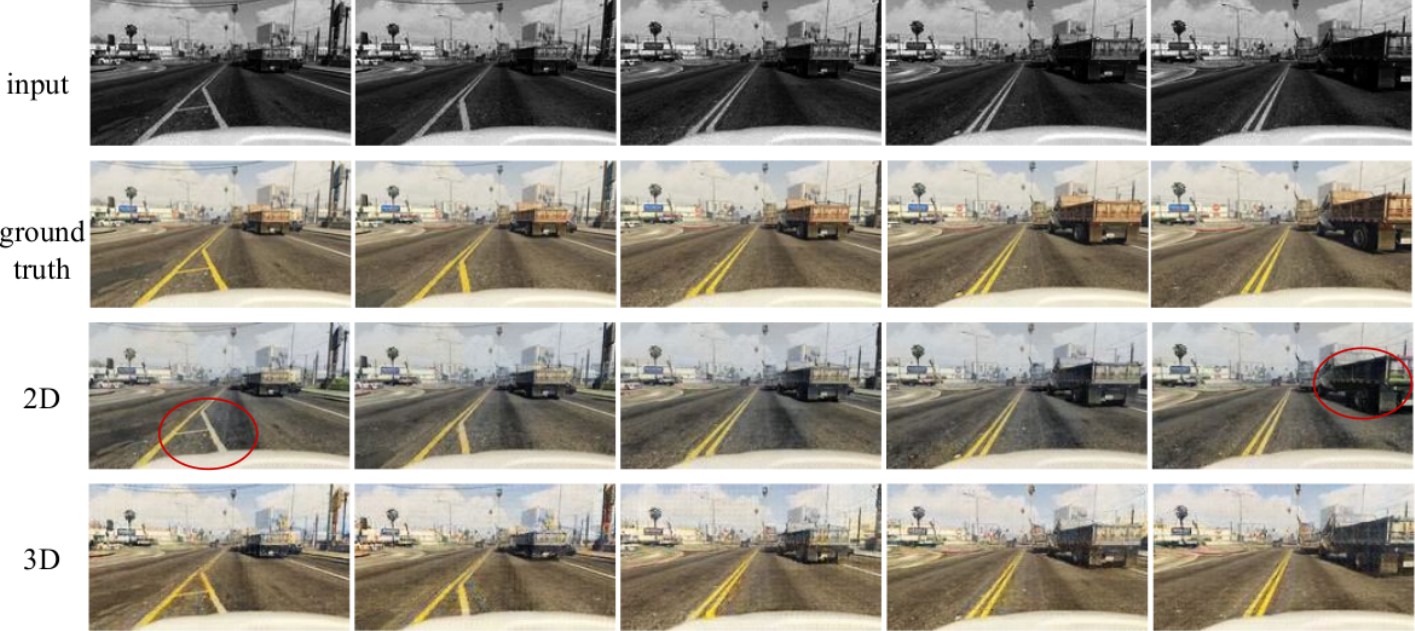

Figure 2: Results of GTA video colorization show that per-frame translation of videos does not preserve constant colours of objects within the whole sequence. We provide more results and videos in the supplementary video: `https://bit.ly/2R5aGgo`. *Best viewed in color.*

Our extensive experiments show that the proposed 3D convolutional model provides more accurate and stable video-to-video translation compared to framewise translation with various settings. We also investigate how the structure of individual batches affects the training of framewise translation models, and find that structure of a batch is very important for stable translation contrary to an established practice of shuffling training data to avoid overfitting in deep models (Goodfellow et al., 2016).

To summarize, we make the following main contributions: 1) a new unsupervised video-to-video translation task together with both realistic and synthetic proof-of-concept datasets; 2) a spatio-temporal video translation model based on a 3D convnet that outperforms per-frame methods in

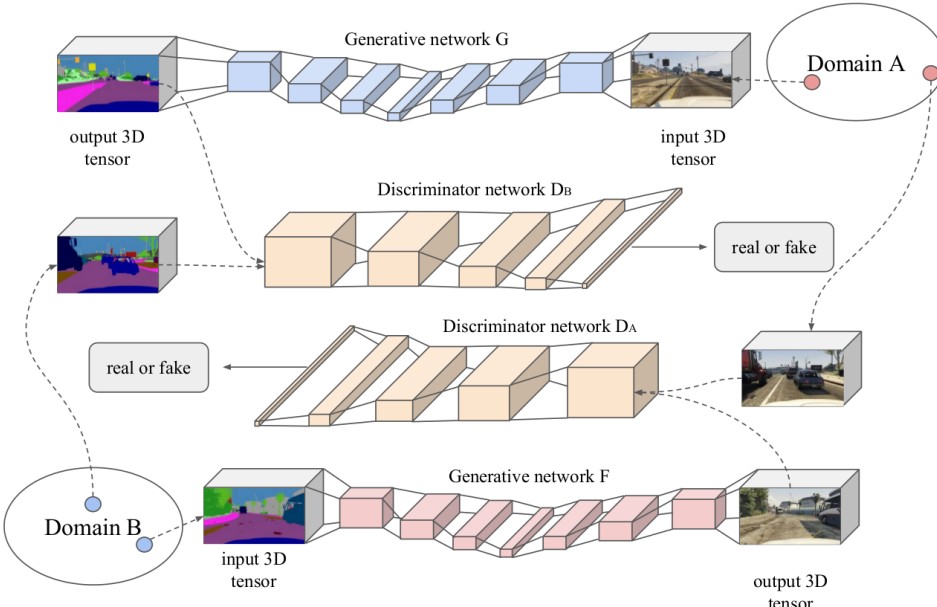

Figure 3: Our model consists of two generator networks ($F$ and $G$) that learn to translate input volumetric images from one domain to another, and two discriminator networks ($D_A$ and $D_B$) that aim to distinguish between real and fake inputs. Additional cycle consistency property requires that the result of translation to the other domain and back is equal to the input video, $G(F(x)) \approx x$.

all experiments, according to human and automatic metrics, and 3) an additional analysis of how performance of per-frame methods depends on the structure of training batches.

## 2 RELATED WORK

In recent years, there has been increasing interest in unsupervised image-to-image translation. Aside from producing interesting graphics effects, it enables task-independent domain adaptation and unsupervised learning of per-pixel labels. Many recent translation models (Zhu et al., 2017; Liu and Tuzel, 2016; Liu et al., 2017b) use the adversarial formulation initially proposed by (Goodfellow et al., 2014) as an objective for training generative probabilistic models. The main intuition behind an adversarial approach to domain translation is that the learned cross-domain mapping $F : X \rightarrow Y$ should translate source samples to fake target samples that are indistinguishable from actual target samples, in the sense that no discriminator from a fixed hypothesis space $\mathcal{H}$ should be capable of distinguishing them.

Many recent advances in domain translation are due to the introduction of the cycle-consistency idea (Zhu et al., 2017). Models with a cycle consistency loss aim to learn two mappings $F(x)$ and $G(y)$ such that not only are the translated source samples $F(x)$ indistinguishable from the target, and $G(y)$ are indistinguishable from source, but they are also inverses of each other, *i.e.* $F(G(y)) = y, G(F(x)) = x$. We also employ this idea and explore how well it generalizes to video-to-video translation. The cycle-consistency constraints might not restrict semantic changes as explored by Hoffman et al. (2017). There has been work on combining cycle-consistency with variational autoencoders that share latent space for source and target (Liu et al., 2017a), which resulted in more visually pleasing results.

Both adversarial (Isola et al., 2016) and non-adversarial (Chen and Koltun, 2017) *supervised* image-to-image translation models archive much better visual fidelity since samples in source and target datasets are paired, but their use is limited since we rarely have access to such aligned datasets for different domains. Adversarial video generation has also gained much traction over last years with frame-level models based on long short-term memory (Srivastava et al., 2015) as well as spatio-temporal convolutions (Vondrick et al., 2016), especially since adversarial terms seem to better avoid frame blurring (Mathieu et al., 2015) than Euclidian distance. However, none of these works consider learning a conditional generative video model from unpaired data, i.e. cross domain video translation. Here, we propose this problem and a solution based on jointly translating the entire volume of video.

## 3 3D CONVOLUTIONAL VIDEO-TO-VIDEO TRANSLATION

We introduce a neural approach for video-to-video translation based on a conditional GAN (Isola et al., 2017; Goodfellow et al., 2014) that treats inputs and outputs as three-dimensional tensors. The network takes a volumetric image (e.g. a video) from domain $A$ and produces a corresponding volume of the same shape in the domain $B$. The generator module aims to generate realistic volumes, while the discriminator aims to discriminate between the real and generated samples. Similarly to the CycleGAN method, we introduce two generator-discriminator pairs and add a cycle consistency loss ensuring that samples mapped from $A$ to $B$ and back to $A$ are close to the originals.

We implement the two generators, $F$ and $G$, as 3D convolutional networks (Ji et al., 2013) that follow the architecture described in (Johnson et al., 2016). The networks consist of three convolutional layers with 3D convolutional filters of shape $3 \times 3 \times 3$, nine resudual blocks and two additional convolutional layers with stride $\frac{1}{2}$. The networks receive image sequences as 3D tensors of shape $d \times h \times w$, where $d$ is the length of sequence and $h$ and $w$ are image height and width if an input video is grayscale, and 4D tensor of shape $d \times h \times w \times 3$ if an input video is colored and represented as a sequence of RGB images. Since 3D convolutional networks require a large amount of GPU memory, the choice of the depth $d$ of the input video is usually limited by the the memory of a single GPU unit; we used $d = 8, h = 108, w = 192$ for the experiments with GTA datasets and $d = 30, h = 84, w = 84$ for all experiments on MNIST. The discriminators $D_A$ and $D_B$ are PatchGANs (Isola et al., 2017) as in CycleGAN, but with 3D convolutional filters. They each receive a video of size $d \times h \times w$ and classify whether the overlapping video patches are real samples from the respective domain or are created by the generator network.

The overall objective of the model consists of the adversarial loss $L_{GAN}$ and the cycle consistency loss $L_{cyc}$. The adversarial $L_{GAN}$ loss forces both the generator networks to produce realistic videos and the discriminators to distinguish between real and fake samples from the domain in a min-max fashion, whereas $L_{cyc}$ ensures that each sample $x \sim p_A$ translated into domain $B$ and back is equal to the original and vice versa, i.e. $G(F(x)) \approx x$ (see Fig. 3).

The adversarial loss $L_{GAN}$ is a log-likelihood of correct classification between real and synthesized volumes:

$$L_{GAN}(D_B, G, X, Y) = \mathbb{E}_{y \sim p_B} \log(D_B(y)) + \mathbb{E}_{x \sim p_A} \log(1 - D_B(G(x))$$

where the generator $G$ is learned in a way that minimizes $L_{GAN}$, while the discriminator $D_B$ aims to maximize it. The cycle consistency loss (Zhu et al., 2017) is the $L_1$ loss between the input volume and result of the reverse translation:

$$L_{cyc} = \mathbb{E}_{x \sim p_A}(\|G(F(x)) - x\|_1) + \mathbb{E}_{y \sim p_b}(\|F(G(y)) - y\|_1)$$

The total objective can be written as follows:

$$\mathcal{L}(G, F, D_A, D_B) = L_{GAN}(G, D_B, X, Y) + L_{GAN}(F, D_A, Y, X) + \gamma L_{cyc}(G, F) \tag{1}$$

Because we employ the cycle loss and the PatchGAN architecture also employed by CycleGAN, we refer to our model as *3D CycleGAN*. More generally, we can consider other generator and discriminator implementations within the overall 3D convolutional framework for video-to-video translation.

## 4 FRAMEWISE BASELINES

We used CycleGAN trained on randomly selected images (referred to as *random CycleGAN*) as a baseline method. We also considered two alternative training strategies for training frame-level CycleGAN baselines: CycleGAN trained on consecutive image sequences (*sequential CycleGAN*) and sequential CycleGAN with additional total variation loss (see Eq. 2) that penalizes radical change in the generated image sequence (*const-loss CycleGAN*). We compared the performance of these baselines with our approach that operates on three-dimensional inputs (*3D CycleGAN*).

**Random CycleGAN.** The first strategy for training a CycleGAN is taking as an input 2D images selected randomly from image sequences available for training, which is the standard approach in deep learning. Data shuffling is known to reduce overfitting and speeds up the learning process (Goodfellow et al., 2016).

**Sequential CycleGAN.** Since the order of frames is essential in sequential image data, we investigated the case when images are given to a 2D CycleGAN sequentially during the training phase (see

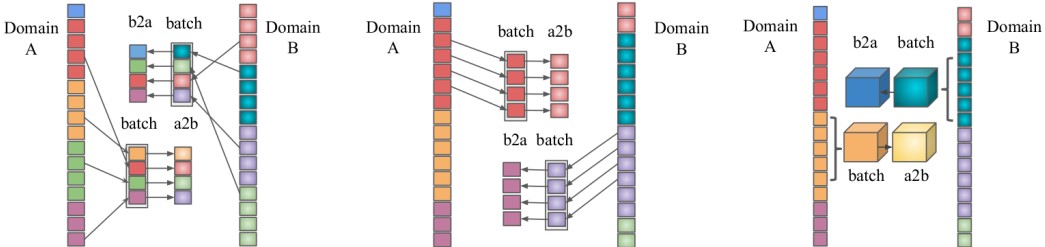

Figure 4: We compare three ways of forming batches used during **training** of a CycleGAN : (a) **random** frames from multiple videos, (b) **sequential** frames from a single video or (c) single **3D** tensor consisting of consecutive frames from a single video. Contrary to the conventional wisdom, our experiments suggest that additional randomness in the batch structure induced in case (a) hurts the performance and convergence of the resulting translation model.

Fig. 4). In contrast to our expectation and the conventional wisdom, sequential CycleGAN often outperformed random CycleGAN in terms of image continuity and frame-wise translation quality.

**Const-loss CycleGAN.** Since CycleGAN performs translation on the frame-level, it is not able to use information from previous frames and, as a result, produces some undesired motion artifacts, such as too rapid change of object shape in consecutive frames, change of color or disappearing objects. To alleviate this problem, we tried to force the frame-level model to generate more consistent image sequences by directly adding a total variation penalty term to the loss function, as follows:

$$L_{const}(G, X) = \mathbb{E}_{x \in X} \sum_t \|G(x)_t - G(x)_{t-1}\|_2^2. \tag{2}$$

## 5 EXPERIMENTS

### 5.1 DATASETS

Intuitively, translation models that operate on individual frames can not preserve continuity along the time dimension, which may result in radical and inconsistent changes in shape, color and texture. In order to show this empirically, we used the GTA segmentation dataset (Richter et al., 2016) for unsupervised segmentation-to-video translation. Since this task is very challenging even for still images, we created three more datasets that give more insight into pros and cons of different models.

**MRCT dataset.** First, we evaluated the performance of our method on an MR (magnetic resonance) to CT (computed tomography) volumetric image translation task. We collected 225 volumetric MR images from LGG-1p19qDeletion dataset (Akkus et al., 2017) and 234 CT volumetric images from Head-Neck-PET-CT dataset (Vallières et al., 2017). Both datasets are licensed with Creative Commons Attribution 3.0 Unported License. Since images from these datasets represent different parts of the body, we chose parts of volume where body regions represented in the images overlap: from superciliary arches to lower jaw. Images of both modalities were manually cropped and resized to $30 \times 256 \times 256$ shape. The final dataset is available for download on the website [TDB].

**Volumetric MNIST.** Volumetric MNIST dataset was created using MNIST handwritten digits database (LeCun, 1998). From each image from MNIST we created a volumetric image imitating 3d scan domain using erosion transformation of two types, we called them "spherical" and "sandglass" domains (see Figure 6). The task is to capture and translate the global intensity pattern over time while preserving image content (digit). The resulting image volumes of shape $30 \times 84 \times 84$ were used to train the models to transform digits of spherical type to sandglass type and vice versa.

**Colorization of moving MNIST digits**. To test models' ability to preserve local information about the color of object, inspired by the Moving MNIST dataset introduced in (Srivastava et al., 2015), we generated a dataset with moving digits of different colors. We used this dataset to train the models to translate from the original moving white digits to the same moving digits in color.

**GTA segmentation dataset.** The Playing for Benchmarks dataset (Srivastava et al., 2015) is a large collection of GTA gameplay recordings with a rich set of available annotations, and currently it is one of the default datasets for evaluation of image-to-image translation methods. Using the daylight driving and walking subsets [1] of this dataset we generated 1216 short clips of shape $30 \times 192 \times 108$ and corresponding ground truth segmentation videos. Since this paper is focused on translation of

---

[1] sections 001-004, 044-049, 051, 066-069

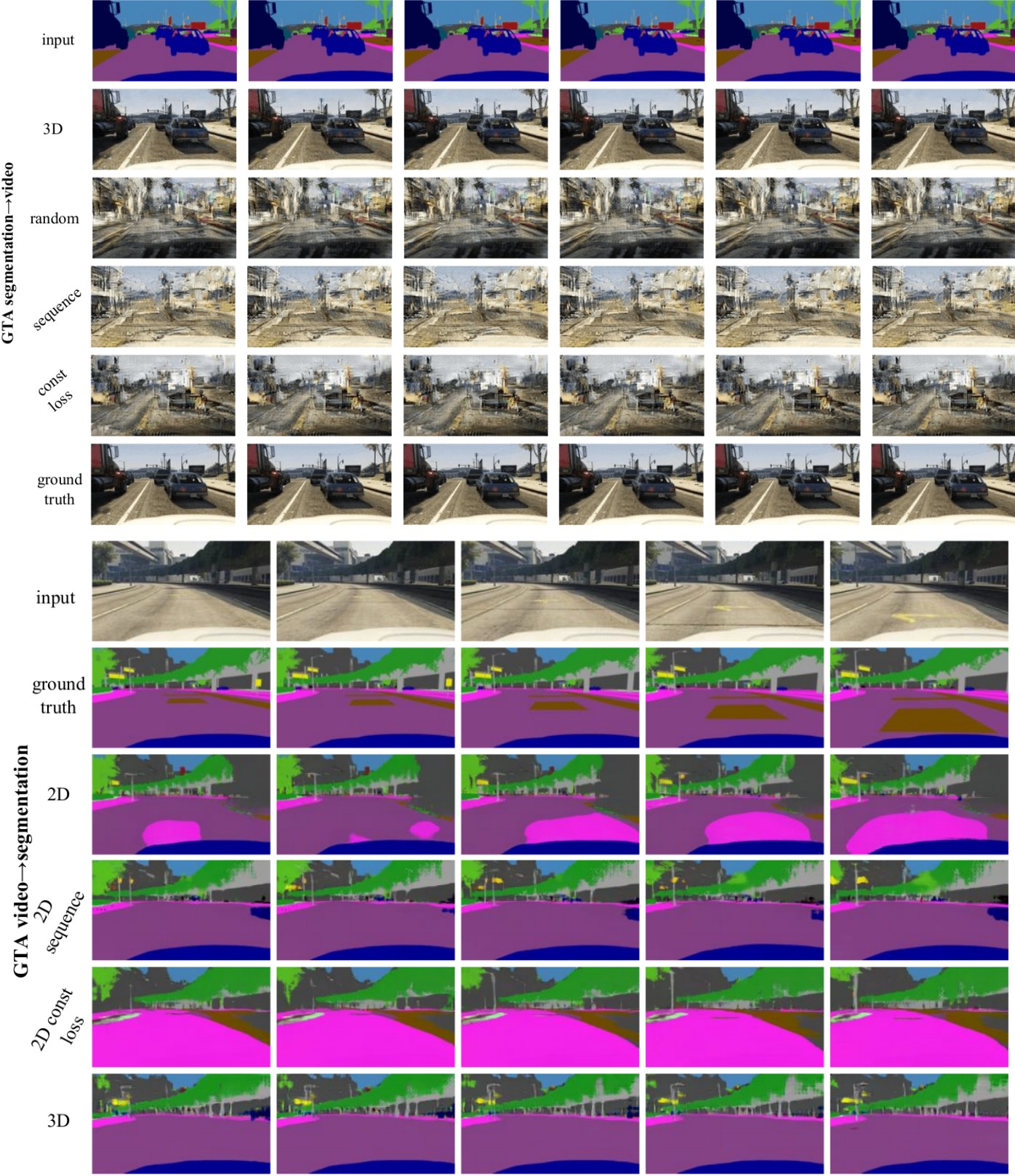

Figure 5: Results of unsupervised GTA video-to-segmentation translation with different models. We observed that frame-level methods diverged significantly more frequently than 3D model (top). No information about ground truth pairs was used during training. Frame-wise translation (2D) produces plausible images, but does not preserve temporal consistency. Forming batches from consecutive frames during training (2D sequence) helps in reducing spatio-temporal artifacts and improves convergence. Additional penalty term on consecutive generated frames (const loss) further reduces motion artifacts at the expense of diversity of generated images. Our proposed 3D convolutional model (3D) produces outputs that are coherent in time, but have fewer details because network has to approximate a higher dimensional mapping (video-to-video instead of frame-to-frame) using same number of learnable weights.

**Volumetric MNIST**

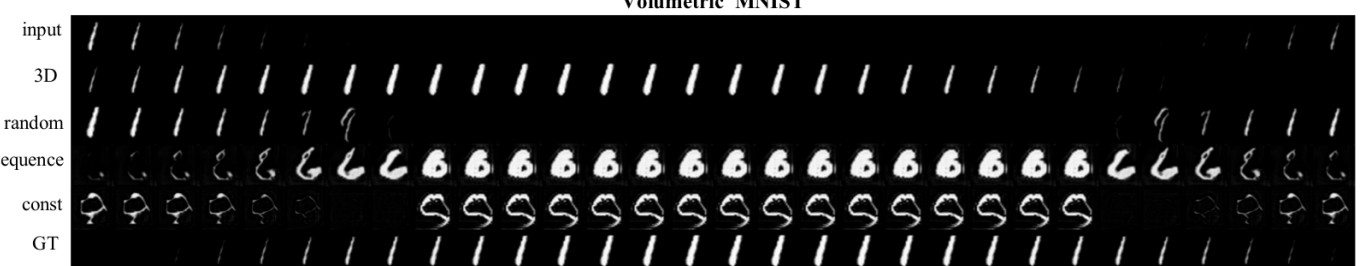

Figure 6: The results of experiments on Volumetric MNSIT dataset: input and output domains contain videos with global decay-then-rise and rise-then-decay intensity patterns respectively, during models were presented with pairs of videos containing *different digits*. Our experiments show that frame-level approaches are not able to learn this spatio-temporal pattern and hence cannot perform correct translation whereas our 3D method performs almost perfectly. Both sequence and sequence+const approaches were able to capture temporal pattern but did not learn shape correspondence.

dense image sequences with a lot of inter-frame relations, we used the walking subset of this dataset for evaluation as it has slower camera movements and therefore higher effective frame rate.

## 5.2 EVALUATION

We performed experiments on all datasets mentioned above with four approaches: random, sequential, sequential+const and 3D CycleGAN. For each dataset we used the same 70% of data for training and 30% for performance evaluation. For datasets that contain ground truth pairs (GTA, volumetric MNIST, colored 3D MNIST) samples from both domains were split into train and test sets independently, i.e. often no corresponding pairs of images were present in the training data. We set the number of parameters to be the same for all models ($\sim 90M$). We also trained the models with number of learnable parameters proportional to the input size ($\sim 200M$ for 3D model). Visual fidelity of the video generated by larger models did not improve, whereas segmentation quality increased significantly. We also report the performance of a large ($\sim 200M$) random 2D model for reference.

For MRCT and GTA segmentation-to-RGB tasks, we ran human evaluation on amazon mechanical turk since these is no "gold" translation to compare with (Table 3). In a series of randomized trials, participants were presented with multiple real samples from each domain and then were asked to choose the more realistic one of the outputs of two different models for *same* input. We also estimated the probability of choosing a video generated by each model over a real one, but only report these numbers for the MRCT domain pair since for segmentation-to-RGB they were below significance level. To help evaluate significance of differences in probabilities, we report a bootstrap estimate of the standard deviation of reported probabilities.

For some domain pairs we actually have definitive ground truth answers. For rgb-to-segmentation translation we evaluated segmentation pixel accuracy and $L_2$ distance between stochastic matrices of pixel class transitions between frames for generated and ground truth segmentation videos with different label denoising levels (Table 4). For volumetric MNIST dataset we also computed $L_2$ error (Table 2). Since there is no single correct colorization in the colorization of moving digits task, we evaluated average standard deviation of non-background colors within each generated image. Models are expected to colorize digits without changing their shapes, therefore we also evaluate $L_2$ error between original and translated shapes (Table 1).

## 6 RESULTS

| Method | shape $L_2$ loss | Intensity mean | colour $\sigma$ |
|---|---|---|---|
| GT | – | 175.37 | 32.25 |
| 3D | **0.79** | 204.36 | 45.65 |
| Random | 1.48 | 139.38 | 64.51 |
| Sequence | 0.84 | 205.49 | 53.96 |
| Seq+const | 0.90 | 198.18 | 54.06 |

Table 1: MNIST video colorization: $L_2$ loss between masks of ground truth and translated images, mean intensity of the digit and standard deviation of the digit color intensity.

| Method | $L_2$ | $\sigma$ |
|---|---|---|
| 3D | **27.21** | 5.27 |
| Random | 48.73 | 13.72 |
| Sequence | 41.73 | 11.40 |
| Seq+const | 73.41 | 5.26 |

Table 2: Volumetric MNIST: $L_2$ loss between the translation (rows 2-5 on figure 6) and ground truth videos (last row).

| Method | $P(A \prec B)$ | | | $P(A \prec GT)$ | |
|---|---|---|---|---|---|
| | GTA | CT-MR | MR-CT | CT-MR | MR-CT |
| 3D | **0.70** | **0.83** | 0.67 | **0.42** | 0.39 |
| Random | 0.21 | 0.58 | 0.65 | 0.24 | 0.44 |
| Sequence | 0.50 | 0.30 | 0.32 | 0.08 | 0.16 |
| Seq+const | 0.59 | 0.29 | 0.36 | 0.07 | 0.21 |
| $2 \times \sigma$ | 0.04 | 0.02 | 0.04 | 0.02 | 0.03 |

Table 3: Human evaluation of generated GTA segmentation-to-video, CT-to-MRI and MRI-to-CT translations. We report probabilities of users preferring each method over other methods, and probabilities of preferring a method over ground truth (below statistical significance for GTA). Last row shows bootstrap estimate of the variance and is similar among all methods.

| Method | Slow | All-Test |
|---|---|---|
| 3D 200M | **0.77** | **0.72** |
| 2D 200M | 0.60 | 0.70 |
| 3D | 0.54 | 0.55 |
| Random | 0.46 | 0.65 |
| Sequence | 0.64 | 0.71 |
| Seq+const | 0.56 | 0.68 |

Table 4: Per-frame pixel accuracy of video-to-segmentation mapping. All proposed methods outperformed the frame-level baseline (random) on both slower and faster videos. With number of learnable parameters proportional to the size of input, 3D model outperformed all other methods.

**Volumetric MNIST.** Our experiments on volumetric MNIST show that standard CycleGAN is not able to capture the global motion patterns in the image sequence (see Fig. 6). Since videos from domain $B$ have the same frames as videos from domain $B$ but in different order, a model with random batches cannot this temporal pattern and outputs a slightly dimmed input. In contrast, 3D CycleGAN was able to learn the transformation almost perfectly. In contrast, sequential models learned the global phase pattern properly, but were unable to generate correct shapes.

The experiment on **colorization of MNIST videos** showed that the "random" model is able to colorize individual frames but cannot preserve the color throughout the whole sequence. The choice of batch selection, however, is important: the sequential and const-loss models learned to preserve the same color throughout the sequence even though they did not have access to previous frames. However, we should mention that all models that succeeded in this task collapsed to colorizing digits with a single (blue-green) color even though the training data had 20 different colors.

The **GTA segmentation-to-video** translation with the 3D model produced smoother and more consistent videos compared to the framewise methods which produced undesirable artifacts such as shadow flickering and rapid deformation or disappearance of objects and road marking. Both sequence methods often moved static objects like road marking with camera. One of the drawbacks of the 3D model is that it does not tolerate rapid changes in the input and hence cannot deal with low frame-rate videos. The additional constraint on total variation resulted in better visual fidelity and smoothness, but leaned towards rendering all objects of same semantic class using same texture, which reduces the variability of the outputs and fine details, but at the same time reduces the amount of spatio-temporal artifacts. The qualitative results of **GTA video colorization** confirm that the spatio-temporal model produces more consistent and stable colorization (see Fig. 1).

The experiments on **MRI-to-CT** translation showed that all per-frame translation methods produce image volumes that do not capture the real anatomy (e.g. shape of the skull, nasal path and eyes vary significantly within the neighboring frames), whereas the proposed 3D method gives a continuous and generally more realistic results for both CT-MRI and GTA segmentation-to-video tasks (Table 3). The CT-to-MRI task is harder since it requires "hallucinating" a lot of fine details and on this task 3D model outperformed random with a significant margin (bold numbers). On a simpler MRI-to-CT task random and 3D models performed similarly within the limits of statistical error.

In contrast to the common practice, the sequential batch approach produced more realistic and continuous results compared to the random batch choice. Supposedly this is due to the fact that images within the sequence are more similar than randomly selected images, and hence the magnitude of the sum of gradients might be higher resulting in faster convergence. Of course, order of frames *within* sequential batch does not matter since all gradients are summed up during backward pass, but the similarity between images within a batch is important.

## 7 CONCLUSION

We proposed a new computer vision task of unsupervised video-to-video translation as well as datasets, metrics and multiple baselines: multiple approaches to framewise translation using image-to-image CycleGAN and its spatio-temporal extension 3D CycleGAN. The results of exhaustive experiments show that per-frame approaches cannot capture the essential properties of videos, such as global motion patterns and shape and texture consistency of translated objects. However, contrary to the previous practice, sequential batch selection helps to reduce motion artifacts.

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

# 8 SUPPLEMENTARY MATERIAL

**MRI→CT**

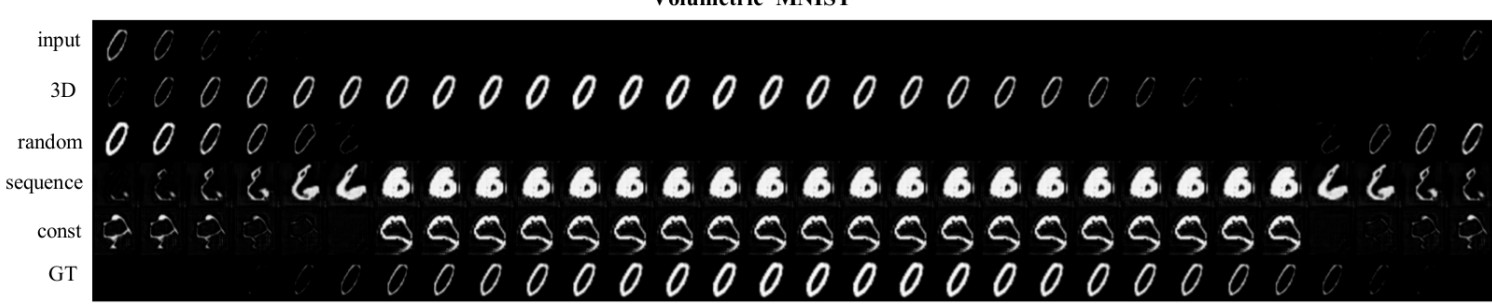

Figure 7: Volumetric MNIST results.

**Volumetric MNIST**

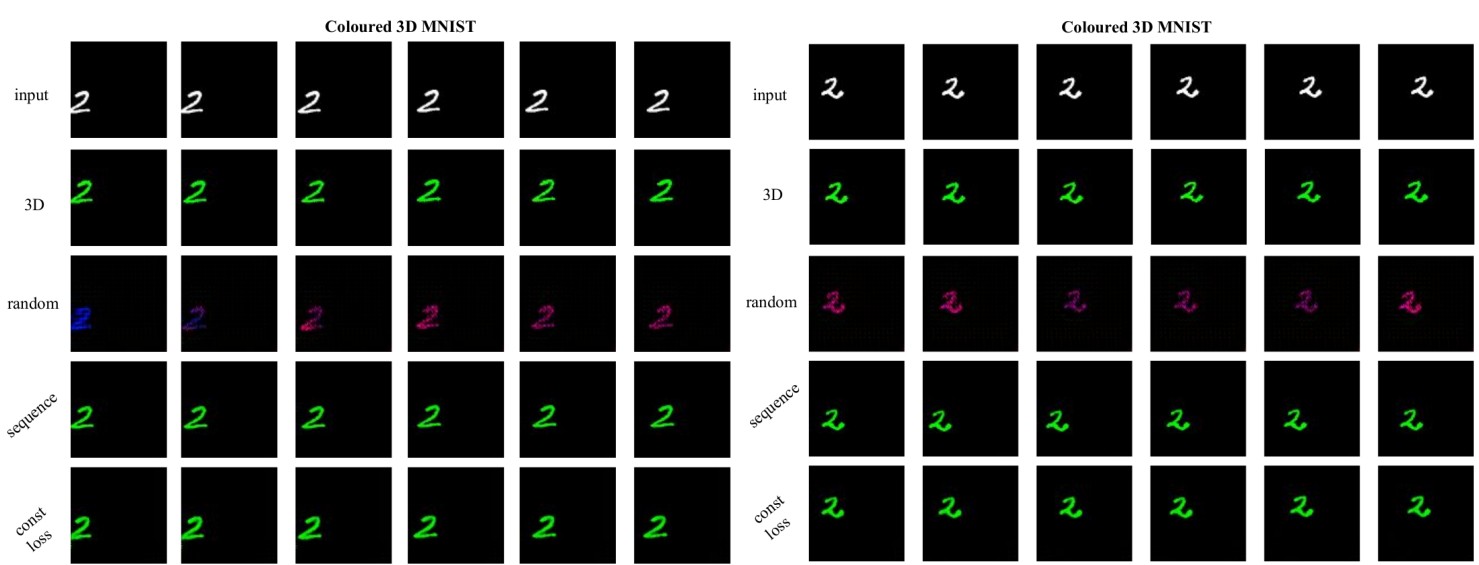

Figure 8: Results of MRI-to-CT translation.

Figure 9: Colored 3D MNIST translation results.

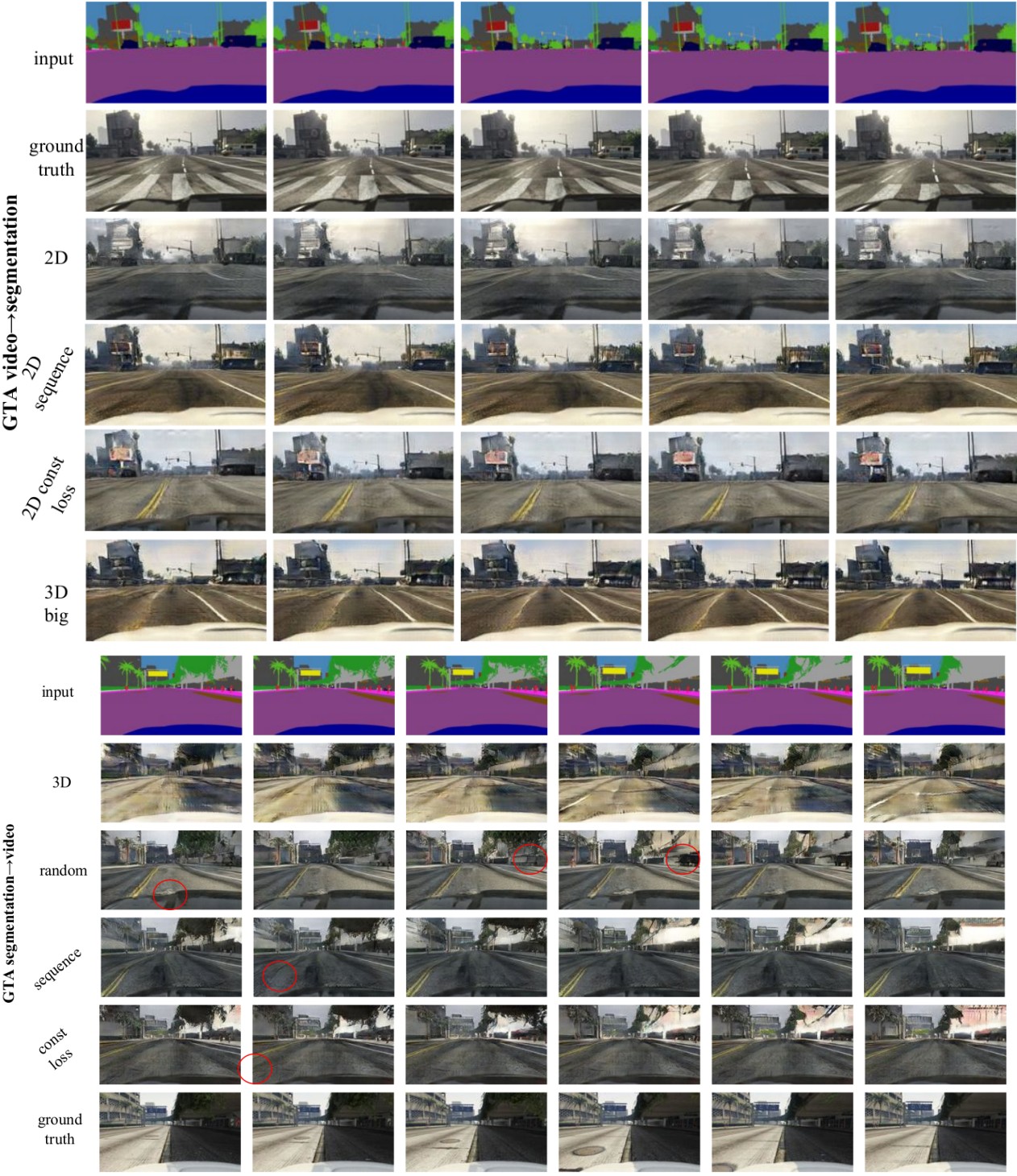

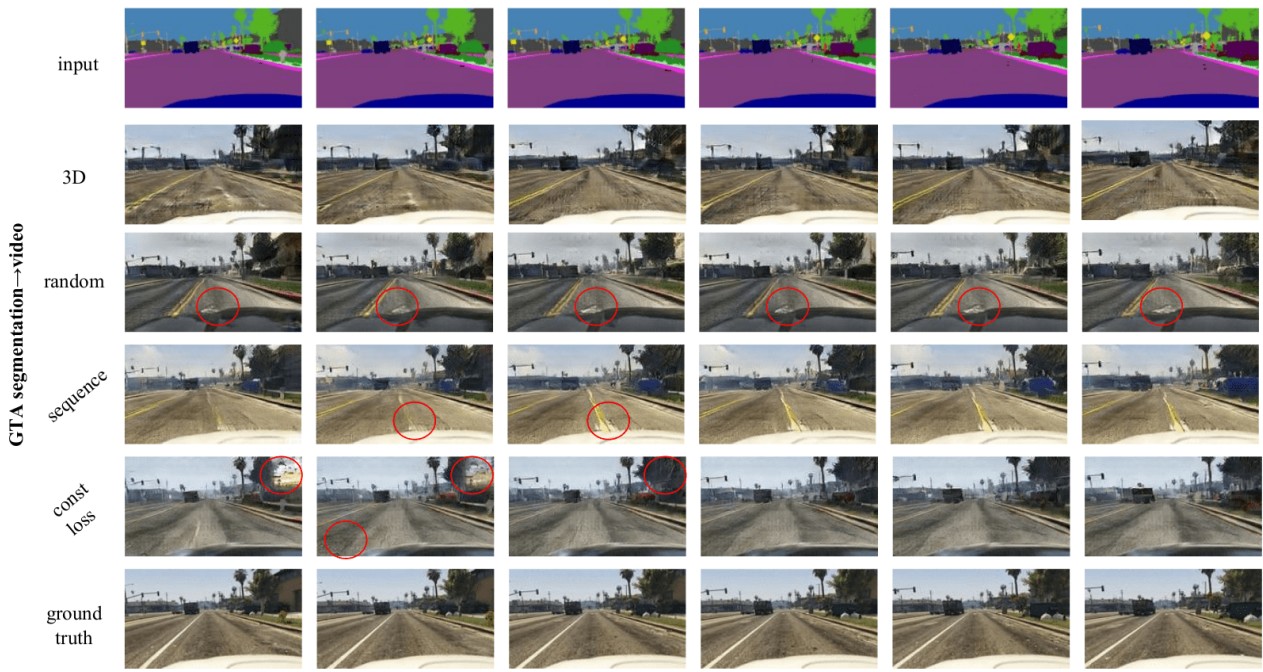