# OpenReview forum: "Unsupervised Video-to-Video Translation"
_ICLR.cc/2019/Conference_

### Official Review · AnonReviewer1 · 2018-10-25
**Limited technical novelty**

**Rating:** 4
**Confidence:** 5

**Review:**

This paper proposes a spatio-temporal 3D translator for the unsupervised image-to-image translation task and a new CT-to-MRI volumetric images translation dataset for evaluation. Results on different datasets show the proposed 3D translator model outperforms per-frame translation model.

Pros:
* The proposed 3D translator can utilize the spatio-temporal information to keep the translation results consistent across time. Both color and shape information are preserved well.
* Extensive evaluation are done on different datasets and the evaluation protocols are designed well. The paper is easy to follow.

Cons:
* The unsupervised video-to-video translation task has been tested by previous per-frame translation model, e.g. CycleGAN and UNIT. Results can be found on their Github project page. Therefore, unsupervised video-to-video translation is not a new task as clarified in the paper, although this paper is one of the pioneers in this task.
* The proposed 3D translator extend the CycleGAN framework to video-to-video translation task with 3D convolution in a straightforward way. The technical novelty of the paper is limited for ICLR.  I think the authors are working on the right direction, but lots of improvement should be done.
* As to Table 4, I am confused about the the per-frame pixel accuracy results. Does the 3D method get lower accuracy than 2D method?
* As to the GTA segmentation->video experiments, the 3D translator seems cause more artifacts than the 2D method (page 11,12). Also, the title of the figure on page 11 should both be “GTA segmentation->video”

Overall, the technical innovation of this paper is limited and the results are not good enough. I vote for rejection.

---

### Official Review · AnonReviewer2 · 2018-11-01

**Rating:** 4
**Confidence:** 5

**Review:**

1) Summary
This paper proposes a 3D convolutional neural network based architecture for video-to-video translation. The method mitigates the inconsistency problem present when image-to-image translation methods are used in the video domain. Additionally, they present a study of ways to better setting up batched for the learning steps during networks optimization for videos, and also, they propose a new MRI-to-CT dataset for medical volumetric image translation. The proposed method outperforms the image-to-image translation methods in most measures.



2) Pros:
+ Proposed network architecture mitigates the pixel color discontinuity issues present in image-to-image translation methods.
+ Proposed a new MRI-to-CT dataset that could be useful for the community to have a benchmark on medical related research papers.

3) Cons:
Limited network architecture:
- The proposed neural network architecture is limited to only generate the number of frames it was trained to generate. Usually, in video generation / translation / prediction we want to be able to produce any length of video. I acknowledge that the network can be re-used to continue generating number of frames that are multiples of what the network was trained to generate, but the authors have not shown this in the provided videos. I would be good if they can provide evidence that this can be done with the proposed network.

Short videos:
- Another limitation that is related to the previously mentioned issue is that the videos are short, which in video-to-video translation, it should not be difficult to generate longer videos. It is hard to conclude that the proposed method will work for large videos from the provided evidence.

Lack of baselines:
- A paper from NVIDIA Research on video-to-video synthesis [1] (including the code)  came out about a month before the ICLR deadline. It would be good if the authors can include comparison with this method in the paper revision. Other papers such as [2, 3] on image-to-image translation are available for comparison. The authors simply say such methods do not work, but show no evidence in the experimental section. I peeked at some of the results in the papers corresponding websites, and the videos look consistent through time. Can the authors comment on this if I am missing something?


Additional comments:
The authors mention in the conclusion that this paper proposes “a new computer vision task or video-to-video translation, as well as, datasets, metrics and multiple baselines”. I am not sure that video-to-video translation is new, as it has been done by the papers I mention above. Maybe I am misunderstanding the statement? If so, please clarify. Additionally, I am not sure how the metrics are new. Human evaluation has been done before, the video colorization evaluation may be somewhat new, but I do not think it will generalize to tasks other than colorization. Again, If I am misunderstanding the statement, please let me know in the rebuttal.



4) Conclusion
The problem tackled is a difficult one, but other papers that are not included in experiments have been tested on this task. The proposed dataset can be of great value to the community, and is a clearly important piece of this paper. I am willing to change the current score if they authors are able to address the issues mentioned above.


References:
[1] Ting-Chun Wang, Ming-Yu Liu, Jun-Yan Zhu, Guilin Liu, Andrew Tao, Jan Kautz, and Bryan Catanzaro. "Video-to-Video Synthesis". In NIPS, 2018.
[2] Xun Huang, Ming-Yu Liu, Serge Belongie, Jan Kautz. Multimodal Unsupervised Image-to-Image Translation. In ECCV, 2018.
[3] Ming-Yu Liu, Thomas Breuel, Jan Kautz. Unsupervised Image-to-Image Translation Networks. In NIPS, 2017.

---

### Official Review · AnonReviewer3 · 2018-11-02
**limited novelty**

**Rating:** 3
**Confidence:** 4

**Review:**

This paper present a spatio-temporal (i.e., 3D version) of Cycle-Consistent Adversarial Networks (CycleGAN) for unsupervised video-to-video translation. The evaluations on multiple datasets show the proposed model is better able to work for video translation in terms of image continuity and frame-wise translation quality.

The major contribution of this paper is extending the existing CycleGAN model from image-to-image translation and video-to-video translation using 3D convolutional networks, while it additionally proposes a total penalty term to the loss function. So I mainly concern that such contribution might be not enough for the ICLR quality.

---

### Meta-Review · Area_Chair1 · 2018-12-04
**Reasonable extension of prior work to additional dimensions.**

**Confidence:** 4
**Recommendation:** Reject

**Metareview:**

In this work, a central idea introduced by CycleGAN is extended from 2D convolutions to 3D convolutions to ensure better consistency of style transfer across time. Authors demonstrate improvements on a variety of datasets in comparison to frame-by-frame style transfer.

Reviewer Pros:
+ Seems to be effective at enforcing improved consistency over time
+ Proposed medical dataset may be good contribution to community.
+ Good quality evaluation

Reviewer Cons:
- All reviewers felt the technical novelty was low.
- Some questions arose around quantitative results, left unanswered by authors.
- Experiments missing some baseline approaches
- Architecture limited to fixed length video segments

Reviewer consensus is to reject. Authors are encouraged to continue their work and take into account suggestions made by reviewers, including adding additional comparison baselines